

# Is type-D personality trait(s) or state? An examination of type-D temporal stability in older Israeli adults in the community

Ada H. Zohar

Clinical Psychology, Ruppin Academic Center, Israel

## ABSTRACT

**Background.** Type D personality was suggested as a marker of poorer prognosis for patients of cardiovascular disease. It is defined by having a score of 10 or more on both sub-scales of the DS14 questionnaire, Social Inhibition (SI) and Negative Affectivity (NA). As Type D was designed to predict risk, its temporal stability is of prime importance.
**Methods.** Participants in the current study were 285 community volunteers, who completed the DS14, and other personality scales, at a mean interval of six years.
**Results.** The prevalence of Type D did not change. The component traits of Type D showed rank order stability. Type D caseness temporal stability was improved by using the sub-scales product as a criterion. Logistic hierarchical regression predicting Type D classification from Time1 demonstrated that the best predictors were Time1 scores on NA and SI, with the character trait of Cooperation, and the alexithymia score adding some predictive power.
**Conclusions.** The temporal stability of the component traits, and of the prevalence of Type D were excellent. Temporal stability of Type D caseness may be improved by using a product threshold, rather than the current rule. Research is required in order to formulate the optimal timing for Type D measurement for predictive purposes.

Type D "distressed" personality type is characterized by high negative affectivity, coupled with elevated social inhibition, making the Type D person unable to gain adequate social support for the weight of negative affectivity he or she experiences (*Denollet, 2005*). Individuals with Type D personality are more likely than others to suffer from social anxiety (*Kupper & Denollet, 2014*). Measured by a self-report questionnaire (DS14; *Denollet, 2005*) Type D is classified when respondents score 10 or more on each of the component traits, social inhibition (SI) and negative affectivity (NA). Type D personality has been found to be a potent risk factor for hypertension and for cardiac vascular disease (*Strike & Steptoe, 2005*). One mechanism putting Type D individuals at risk is thought to be that high levels of chronic stress lead to high concentrations of stress hormones, harming the membranes of blood vessels and allowing the build-up of plaque, which in turn raises blood pressure and makes cardiac events more likely. There is some proof of causality; not only does Type D personality raise the probability of cardiac vascular disease (CVD), but addressing the distress of Type D personality patients after a cardiac event

Corresponding author
Ada H. Zohar, adaz@ruppin.ac.il

leads to significantly lower mortality and morbidity (*Denollet & Brutsaert, 2001*). Another possible mechanism is that Type-D individuals may engage in less healthy behavior. A study of patients with heart failure in the United States (*Wu & Moser, 2013*) found that Type D patients were less likely to adhere to their medication. Indonesian coronary heart patients who were Type D engaged less in health behavior than Non-D patients (*Ginting et al., 2014*). A study of Dutch patients attending an outpatient cardiac clinic (*Schiffer et al., 2005*) showed that Type D personality tripled the risk of heart failure and increased the risk for depressive symptoms more than six-fold. A longitudinal study of over 500 cardiac patients (*Denollet et al., 2013*) found significant odds ratio for Type D cardiac patients to suffer a major cardiac event (MACE), i.e., a myocardial infarction, coronary revasculation, or cardiac death. These effects did not hold when using the component traits of Type D, SI and NA as continuous risk factors and depended on the interaction, i.e., both traits being above a cut-off of 10. A meta-analysis showed Type D to confer additional risk or poorer prognosis for CVD patients (*Grande, Romppel & Barth, 2012*). It should be noted that in some studies, depression is found to be a better prognostic predictor than Type D status (e.g., *Damen et al., 2013*). In others, Type D is not associated with poorer health behavior (*Habibović et al., 2014*).

These findings about Type D show a strong effect, but the question whether or not Type D personality is a discrete entity has yet to be addressed empirically. The features associated with Type D, negative affectivity and social inhibition, can arise from multiple continuous traits with different psychological and biological causes. Moreover, if Type D is a discrete entity, one would expect it to have high temporal stability.

Temporal stability has been variously defined. In a large-scale study of healthy Dutch twins, *Kupper et al. (2011)* showed that Type D itself and both of its component traits had stable genetic influences which did not change over the 9 year study. In trait-personality models, temporal stability is usually reported as the correlation of trait scores at different time-points, often called rank-order stability. This can be applied in the current study to the component traits of Type D, NA and SI. However, this measure misses the essence of Type D, i.e., that it is a dichotomous classification, and thus very different from most current personality models. To measure temporal stability of the Type D dichotomous classification, two additional measures of temporal stability were considered: (1) the prevalence of Type D individuals at both time-points, and (2) The proportion of individuals who were classified as Type D at T1 who still qualified for Type D at T2; and the proportion of T1 non-D individuals who still qualified for Non-D status at T2. Since Type D is defined by an absolute threshold (a score of 10 or more on both of the subscales), it was also possible to examine the temporal stability of the dichotomous classification, by examining an alternate definition of "Dness": examining the criteria points provided by the product of NA and SI scores for temporal stability of this alternate classification.

The current study addresses the question of temporal stability of Type D in a non-clinical sample of Israeli adult community volunteers, measured twice at a mean interval of six years. At outset, 1,350 volunteers completed the DS14 (*Zohar et al., 2011*). At that time, 24.1% of the participants were Type D positive, and on average the Type D group differed substantially from the Non-D group: they were more alexithymic, reported

poorer subjective health, less social support and lower satisfaction with life. There was also a significant association between being Type-D positive and having a known medical diagnosis of CVD or diabetes. Individuals who were Type–D positive were significantly different from the Non-D individuals on six of the seven Temperament and Character (TCI) traits: more harm avoidant, less novelty seeking, less reward dependent, less persistent, less self-directed and cooperative. A mean six years later about a quarter of the original sample were available for re-testing.

This study wished to examine the following questions: 1. The temporal stability of Type D prevalence. 2. The rank-order stability of the component traits of Type D, SI and NA. 3. The temporal stability of the Type D classification using the accepted criterion. 4. Examining as threshold points the product of the sub-scale scores of the DS14 to see if they provide more stable classifications than the Type D membership. 5. Using the extensive personality scales used at Time1 and described in detail elsewhere (*Zohar et al., 2011*) to add to the prediction of Time2 Type D classification.

## METHOD

### Participants

The participants were 285 community volunteers, enrolled in a longitudinal study of personality and health. The baseline sample is described in detail elsewhere (*Zohar & Cloninger, 2011*). In the current study only those participants who had previously agreed to take part in the longitudinal study, and who were still alive, and who had access to the internet were contacted. This included 471 potential participants. Comparing this subset to the original baseline sample on all personality and demographic variables showed that this subset did not differ from the baseline sample in any of the variables except mean age—this sample was on average about two years younger. Of those 471 contacted, 60.1 %, or 285, completed the extensive on-line self-report which is the time 2 (T2) data. Of these, 42.4% were men. The participants' age ranged between 45 and 95, with a mean of 62.2. Their education ranged between partial primary school and PhD with a mean of 15.75 years of education, i.e., college education. Most, 68.4% were married, 19.2% were divorced, and 8.4% were widowed. The final sample was comparable to the baseline sample on all personality measures: the TCI traits, Type D traits, and alexithymia.

### Procedure

The baseline measurements were reviewed and approved by the ethics committee of the neighboring hospital, approval #42/2007. The second time point measurements were reviewed by the institutional IRB, who also approved the electronic informed consent procedure for the online self-report. Potential baseline participants were contacted by email, and those who agreed to participate in this phase of the study were mailed a link to an online questionnaire.

### Measures

#### DS14 (*Denollet, 2005*)

This 14 item questionnaire includes 7 items which measure negative affectivity (NA) and 7 items which measure social inhibition (SI). Type D personality is confirmed when

an individual scores 10 or more on both the subscales. The DS14 was found to perform very well in Hebrew (*Zohar et al., 2011*). It showed structural validity in exploratory and confirmatory factor analyses, and convergent and divergent validity against other personality scales, in particular the temperament and character inventory, (TCI; *Zohar & Cloninger, 2011*).

### TCI-140

This version of the Temperament and Character Inventory includes 140 items which are answered on a 5-value Likert-like scale. It measures four temperament traits: harm avoidance (HA); novelty seeking (NS); reward dependence (RD) and persistence (PS). In addition, it measures 3 character traits: self-directedness (SD); cooperation (CO) and self-transcendence (ST). The TCI-140 performs very well in Hebrew (*Zohar & Cloninger, 2011*).

### Toronto Alexithymia Scale-20 (TAS20)

*Bagby, Parker & Taylor (1994)* constructed a 20 item 5-point response scale for alexithymia which reduces to 3 subscales, difficulty in identifying feelings, difficulty in describing feelings, and externally-oriented thinking. A total score over 61 is considered evidence of alexithymia, and a total score of less than 51 is considered evidence for non-alexithymia (*Taylor, Bagby & Parker, 1997*).

## Data analysis

All data were entered directly onto SPSS via self-report using Qualtrics. Hypothesis testing was conducted using SPSS21.0 for WINDOWS. Only complete reports were considered in this study.

## RESULTS

The potential score for each of the 14 items of the DS14 is 0–4. Thus for each of the sub-scales, the scale score ranges from 0 to 28. For SI the mean at T1 was 9.59 (SD = 5.4) and at T2 9.04 (SD = 6.7). For NA the mean at T1 was 8.96 (SD = 5.3) and 8.76 (SD = 5.8) at T2.

## Type D prevalence

In the original base sample ($N = 1,350$; *Zohar et al., 2011*) the prevalence of Type D was found to be 24.1%. A subset of the base sample who completed the T2 evaluation are presented in the current study. Of them 72 or 25.3% were Type D at T1. A mean six years later, 62 of the 285 participants, or 21.4% were Type D. The rates of Type D at the various time-points are not different: Baseline vs. baseline-subset: $X^2 = 0.1589$, $p > 0.05$; baseline-subset vs. Time2: $X^2 = 1.1867$, $p > 0.05$. Thus the **prevalence** of Type D personality is temporally stable.

## Rank-order-stability of DS14 Traits

Rank-order stability of the DS14 subscale scores was assessed by calculating the correlation between the traits at T1 and T2. For SI, the correlation was 0.818, $p < 0.001$. For NA it was 0.723, $p < 0.001$.

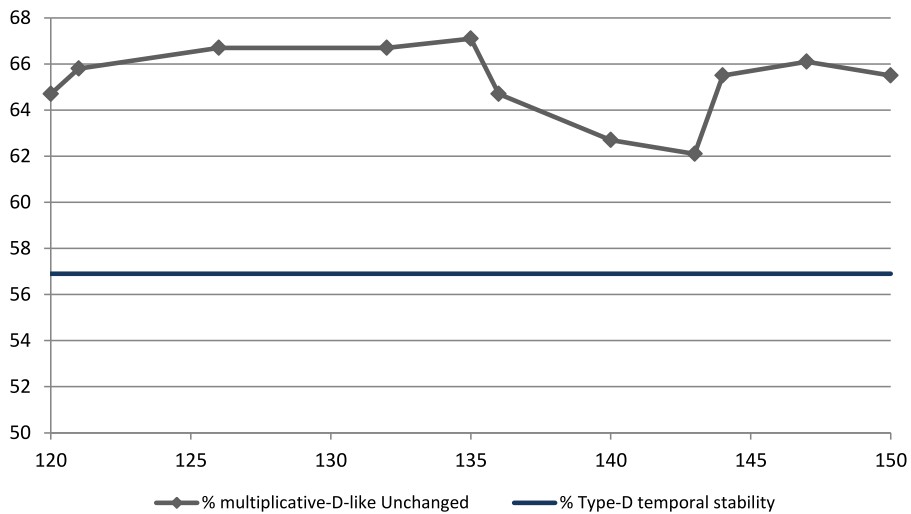

**Figure 1** **Six-year temporal stability of multiplicative-D-like criterion vs. Type D.** Note: on the $x$-axis the product of the DS14 sub-scale scores used as the cutoff point. On the $y$-axis the percentage of T1 multiplicative-D-like individuals who were still multiplicative-D-like at T2. For easy reference, parallel to the $x$-axis, the stability of Type D membership, 56.9%.

## Type D membership temporal stability

Were the same individuals who were classified as Type D at T1 classified as Type D at T2? This question was examined by cross-tabulating the 285 participants for Type D classification at both time-points. The association between the classification at both time-points was strong: $X^2 = 72.34$, $p < 0.001$. However, of the individuals originally qualifying for Type D, $N = 72$, only 41 or 56.9% still qualified for Type D. Of those originally Non-D, $N = 213$, 193 or 90.6% remained Non-D.

## Is the product of the two sub-scale scores more temporally stable than the Type D criterion?

The Type D criterion is essentially non-linear. A non-linear alternative is to consider the product of the two sub-scale scores. This was done by calculating the product of the two sub-scale scores for each participant at each time point. The rank order stability of the product from T1 to T2 was $r_{product} = 0.781$, $p < 0.001$. We set a threshold for multiplicative-D-ness at a value and then checked what proportion of multiplicative-D-like individuals retained their status at the six-year retest. Figure 1 shows the results. The product was 0 for individuals who scored 0 on one of the subscales. It should be noted that by the standard Type D criterion these individuals of course would not qualify for Type D. The maximum possible value for the product is $28^2 = 784$; this maximal score was not found at either time-point for any individual. At T1 there were 4 individuals with a score over 500 and at T2 3 such individuals. The product values considered in the subsequent analyses are 120–150; 70% of the participants had a product of less than 120, and 80% a product less than 150.

**Table 1** Summary of binary logistic regression analysis for T1 personality variables predicting Type D membership at T2 ($n = 285$), controlling for gender and age.

| Predictor | $B$ | $SE\ B$ | $e^B$ |
|---|---|---|---|
| $SI_{T1}$ | .207*** | .06 | 1.23 |
| $NA_{T1}$ | .155** | .05 | 1.17 |
| $NS_{T1}$ | .017 | .03 | 1.02 |
| $HA_{T1}$ | .011 | .03 | 1.01 |
| $RD_{T1}$ | −.02 | .03 | 0.98 |
| $PS_{T1}$ | −.02 | .02 | 0.98 |
| $SD_{T1}$ | .001 | .03 | 1.00 |
| $CO_{T1}$ | −.08* | .03 | 0.93 |
| $ST_{T1}$ | .04 | .02 | 1.043 |
| $TAS20_{T1}$ | −.09* | .05 | 0.86 |
| Constant | 0.76 | | |
| $\chi^2\ (df = 13)$ | 100.23 | | |
| % predicted correctly T2 Type D | 49.2 | | |

**Notes.**

$e^B$, exponentiated $B$ (Odds ratio). The T1 sub-script signifies that variables were measured at Time 1, 6 years prior to second testing of predicted Type D.

SI, Social inhibition (DS14); NA, Negative affectivity (DS14); NS, Novelty seeking (TCI); HA, Harm avoidance (TCI); RD, Reward dependence (TCI); PS, Persistence (TCI); SD, Self-directedness (TCI); CO, Cooperation (TCI); ST, Self-transcendence (TCI); TAS20, Total alexithymia score.

*$p < .05$.
**$p < .01$.
***$p < .001$.

As Fig. 1 shows, the temporal stability of the multiplicative-D-like criterion results in higher stability than Type D membership for all values between 120 and 150. There is a local maximal stability point at product = 135. Using 135 as the threshold, produces a prevalence of 23.6% and 24.9% at T1 and T2 respectively. This multiplicative criterion overlaps Type D membership 81.7% and 85.9% at T1 and T2 respectively. Temporal stability of the product at-risk group for threshold 135 is 67.1%.

## Can Type D membership at T2 be predicted using T1 personality variables?

T2 Type D membership was entered as the dependent variable into logistic regression; independent variables used as predictors were the following T1 variables: the scores on both sub-scales of the DS14 at T1, the seven TCI traits at T1, and the total alexithymia score on the TAS20 at T1. The results of the logistic regression are shown in Table 1.

As shown in Table 1, 49.2% of T2 Type D individuals were correctly predicted by the binary logistic regression equation. Since the ratio between Type D and Non-D is 1:4 the logistic regression showed reasonable predictive accuracy. By far the strongest predictors were DS14 subscale scores at T1, followed by some additional predictive power from the cooperativeness TCI character trait, which was inverse to Type D classification, as was the T1 alexithymia score.

## DISCUSSION

The current study found support for temporal stability of Type D using a variety of approaches. It found temporal stability for the *prevalence* of Type D caseness, so that the prevalence at T1 of the current study sample was 25.3%, not statistically different from the prevalence 6 years later, 21.4%. *Jellesma (2008)* found the prevalence of Type D unchanged over an 18 month span in Dutch schoolchildren; and *Ossola et al. (2015)* found the prevalence of Type D unchanged in intensive care coronary patients in Italy over a 12 month span. Thus in a variey of contexts, Type D prevalence is a stable characteristic of populations, and the prevalence itself of about 1 in 4 seems to be very similar in many if not in all cultures. A study of Type D in Korea (*Lim et al., 2011*) showed a remarkably similar prevalence of Type D in healthy cotrols, patients with hypertension and without CVD, and in CVD patients.

The current study also found rank-order stability for the component traits negative affectivity and social inhibition, 0.73 and 0.82 respectively. These correlations are as high as for many other personality traits measured by self-report, and as high or higher than the rank order stability of NA and SI reported by others e.g., *Martens et al. (2007)* or *Spindler et al. (2009)*.

Also in support of the entity of Type D, are the results of the binary logistic regression analysis. The best predictors of Type D classification at Time2 were the Time1 DS14 trait scores. The original trait scores of social inhibition and negative affectivity, each measured by seven items, did better than the seven TCI traits (each measured by 20 items) and better than the alexithymia scale (again 20 items). Thus there is a particular affective and social style measured by the DS14 which is relatively robust, and does better than other more elaborate personality scales in measuring these traits, as well as predicting the Type D criterion, i.e., both sub-scale scores 10 or above, six years later.

A different but intuitively obvious measure of temporal stability is caseness: what proportion of those classified as Type D at T1 will still be Type D at T2, and what proportion of Non-D will remain Non-D? Is it enough that the Chi-Square value for the association is significant, or do we expect a higher level of temporal stability? The current study found, that six years later, 56.9% of Type D adult community volunteers remained Type D, and 90.6% of those originally Non-D remained Non-D. Comparing this result to others in the extant literature is not trivial, because the research on the temporal stability of Type D is relatively new, and in those studies published, the definition of temporal stability differs between studies.

Some studies of CVD patients, report the percentage of patients who remain Type D at retest. In a Swedish study of CVD patients, over 12 months, 6.1% of individuals were Type D at each of three consecutive testing times (*Condén et al., 2014*); this very low temporal stability may be due to variation in assessment method at the different time-points. In a study of consecutive patients who were admitted into a coronary intensive care unit during a 3-year window, patients were assessed every month for 12 months (*Ossola et al., 2015*). They found that among depressed patients, the initial prevalence of Type D was very high, 60%, and over 12 months, 29.2% retained their initial Type D status. Among

non-depressed patients, the prevalence of Type D was 30%, and at 12 months from initial testing 31.3% retained their Type D status. Thus although the prevalence of Type D was double in the depressed coronary patients vs. the non-depressed, staying true to type was comparable in both groups ~30%. They found that over the 12 months Negative Affectivity was significantly reduced, while Social Isolation was constant over the 12 months of the study. In a German study of CVD patients *Romppel et al. (2012)* followed healthy controls, CVD inpatients and outpatients over a 6-year interval. The overall prevalence of Type D in their study was similar in the various groups, and was about 26%. About half the patients with Type D, 14.4% of the patient sample, retained their Type D status. They reported that there was more change in NA than in SI over the six years, which accounted for most of the change in "caseness." *Jellesma (2008)* studied children 8–12 years of age sampled through Dutch schools at an 18 month interval, and found that 42.9% of baseline Type D children remained Type D, and 79.3% of baseline Non-D children remained Non-D. These results fall between the lower stability found in CVD adult patients, who may experience their illness and recovery as transformative, and lower than that found in the current study, in older adults whose personality has developed and stabilized. Other studies report the proportion of temporal stability of Type D classification overall, combining Type D membership and Non-D membership (e.g., *Pelle et al., 2010*). Since Type D individuals are about 1 in 4 or 1 in 5, the a-priori probability of remaining Type D is much lower than the a-priori probability of remaining Non-D, thus overall temporal stability is not an informative measure for the stability of the at-risk-group Type D.

Compared to clinical samples, the current study showed high temporal stability, as it extends over a considerably longer time frame, and as more than half of the individuals originally identified as Type D remained Type D. However, the temporal stability of Type D classification was significantly lower than that achieved for using the product of the subscale scores as a cutoff point. The product of the subscale scores showed a highly significant overlap with the classical Type D classification, as well as having higher temporal stability.

*Coyne & De Voogd (2012)* claimed that Type D classification has not been borne out by empirical research even though the component traits are of obvious importance, and suggest using alternative approaches to scoring the Type Dness, based on the quasi-continuous subscale scores, rather than using the *Denollet (2005)* rule of 10 or more on both sub-scales. Subsequently, *Denollet et al., (2013)*, conducted a five-year longitudinal study of a series of over 500 CVD patients showed that Type D did better at predicting major cardiac events than did its component traits, making the OR for any major cardiac event 1.74, and for cardiac death 2.35. This impressive study showed very strong "positive" results for Type D's predictive validity, but did not address the question of the temporal stability. Was it the Type D status at the beginning of the study that had this major effect? Was the effect different for those who remained stably Type D throughout the five years of the follow-up? If Type D status is at most 30% temporally stable in CVD patients at what point does being Type D exert its influence? Should being Type D be construed of as a state of heightened risk, or as a trait of individuals at elevated risk?

The current study contributes to existing research on the temporal stability of Type D by examining a community sample and not a CVD patient sample, by using a battery of

well validated personality measures, and by having a relatively long inter-test interval. It suggests an alternative criterion for defining Type D, based on the product of the DS14 subscales, which provides better temporal stability in a non-patient sample. In addition, it should be noted that in this study the original Type D criterion produces higher temporal stability relative to other published results.

The results of this study should be viewed with the study limitations in mind, i.e., they may not generalize well to CVD patients, especially those before and after a traumatic cardiac event or a major intervention. These life-threatening experiences may have a dynamic of their own. The current study has other weaknesses: only 60% of those available for testing at T2 completed the evaluation and were included at T2, and the mode of self-report changed from pencil and paper at T1, to an online survey at T2. It should be noted however that the only variable that differed between the subset of the initial sample and the complete baseline sample was age. No personality or other demographic variables were different, and thus it is unlikely that there was selective attrition relevant to personality.

Type D is designed to predict risk, thus its temporal stability is of prime importance, and the question of when it should be measured, or which of subsequent measurements is crucial to understanding the behavior and medical risk of CVD patients, requires further research.

### Funding
The author received no funding for this work.

### Competing Interests
Ada H. Zohar is an Academic Editor for PeerJ.

### Author Contributions
- Ada H. Zohar conceived and designed the experiments, performed the experiments, analyzed the data, contributed reagents/materials/analysis tools, wrote the paper, prepared figures and/or tables, reviewed drafts of the paper.

### Human Ethics
The following information was supplied relating to ethical approvals (i.e., approving body and any reference numbers):

Time1 was approved by the Helsinki Committee at Hillel Yaffe Hospital Hadera, 42/2007.

Time2 was approved by the Ethics Committee, School of Social and Community Sciences, Ruppin Academic Center, an internal review board.

### Data Availability
The raw data and the syntax from SPSS are supplied as Supplemental Information 1 and 2.

## Supplemental Information

Supplemental information for this article can be found online at http://dx.doi.org/10.7717/peerj.1690#supplemental-information.

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
