# Peer review of "Is type-D personality trait(s) or state? An examination of type-D temporal stability in older Israeli adults in the community"

_PeerJ, doi:10.7717/peerj.1690_

## Round 0.1 · original submission · Major Revisions

· Academic Editor

Major Revisions

Your manuscript received two reviews, both requesting revisions . The process of re-review will be facilitated if you respond to the reviewers on an item-by-item basis and specify where changes have been made (yellow highlighting of revised text would be helpful). Be sure that reviewers' requests are addressed in the text, and not merely in the response to reviewers.

Reviewer 1 ·

Basic reporting

A generally well written article. Some prior literature dealing with the temporal stability of the type D construct is missing (some of the articles only reporting rank-order stability):

Jellesma, F. C. (2008). Health in young people: social inhibition and negative affect and their relationship with self-reported somatic complaints. Journal of Developmental and Behavioral Pediatrics: JDBP, 29(2), 94–100. http://doi.org/10.1097/DBP.0b013e31815f24e1
Kupper, N., Boomsma, D. I., de Geus, E. J. C., Denollet, J., & Willemsen, G. (2011). Nine-year stability of type D personality: contributions of genes and environment. Psychosomatic Medicine, 73(1), 75–82. http://doi.org/10.1097/PSY.0b013e3181fdce54
Lim, H. E., Lee, M.-S., Ko, Y.-H., Park, Y.-M., Joe, S.-H., Kim, Y.-K., … Denollet, J. (2011). Assessment of the type D personality construct in the Korean population: a validation study of the Korean DS14. Journal of Korean Medical Science, 26(1), 116–123. http://doi.org/10.3346/jkms.2011.26.1.116
Ossola, P., De Panfilis, C., Tonna, M., Ardissino, D., & Marchesi, C. (2015). DS14 is more likely to measure depression rather than a personality disposition in patients with acute coronary syndrome. Scandinavian Journal of Psychology, 56(6), 685–692. http://doi.org/10.1111/sjop.12244
Pedersen, S. S., Yagensky, A., Smith, O. R. F., Yagenska, O., Shpak, V., & Denollet, J. (2009). Preliminary evidence for the cross-cultural utility of the type D personality construct in the Ukraine. International Journal of Behavioral Medicine, 16(2), 108–115. http://doi.org/10.1007/s12529-008-9022-4
Romppel, M., Herrmann-Lingen, C., Vesper, J.-M., & Grande, G. (2012). Six year stability of Type-D personality in a German cohort of cardiac patients. Journal of Psychosomatic Research, 72(2), 136–141. http://doi.org/10.1016/j.jpsychores.2011.11.009
Spindler, H., Kruse, C., Zwisler, A.-D., & Pedersen, S. S. (2009). Increased anxiety and depression in Danish cardiac patients with a type D personality: cross-validation of the Type D Scale (DS14). International Journal of Behavioral Medicine, 16(2), 98–107. http://doi.org/10.1007/s12529-009-9037-5
Yu, D. S. F., Thompson, D. R., Yu, C. M., Pedersen, S. S., & Denollet, J. (2010). Validating the Type D personality construct in Chinese patients with coronary heart disease. Journal of Psychosomatic Research, 69(2), 111–118. http://doi.org/10.1016/j.jpsychores.2010.01.014

Experimental design

No comments.

Validity of the findings

line 164: The conclusion "Type-D membership does not bias for or against continued participation" is not supported by the stable prevalence and is at risk for an ecological fallacy. The proper test for this would be a comparison of the type D rates for participants and non-participants (which has already been reported in line 123, I assume).

Reporting the rank-order stability for the sum and the product of the scales would add important evidence, especially with regard to the discussion about using continous subscale scores instead of the dichotomization (see Coyne & deVoogd, 2012).

Additional comments

Some minor remarks:

line 57: missing space in "Schifferet al"

line 64: the meta-analysis by Grande et al. does not include "many smaller studies". There are n=12 studies with sample sizes ranging from n=51 to n=977

line 98: the abbreviation "TCI" should be introduced at this point

line 123: It is not clear what is meant by "personality variables": DS14 and TCI? Or sociodemographic variables too?

line 202: "a cutoff value of is just higher than the minimum stability" is difficult to understand. How can a cutoff value be higher than a stability?

line 244: There is a fragmentary phrase: "Since the ratio...1:4". Could be deleted since the argument is repeated in line 257.

line 274: The wording "11.1% of individuals originally tested as Type-D remained Type-D" is unclear at least (I don't see the difference to "retain their status"). 11.1% is the proportion of all participants that showed Type-D at both T1 and T2.

line 277: "Assessment" instead of "assessment"

Reviewer 2 ·

Basic reporting

Adequate.

Experimental design

Adequate.

Validity of the findings

I am afraid it is not appropriate to calculate a total DS14 score. The DS14 assesses two distinctly different personality traits, NA and SI: therefore, scores on these NA and SI measures should not be added as a total DS14 score. Accordingly, it is not appropriate to use cut-off points on the total score of the DS14 to assess the temporal stability of the Type D construct. Sections using total DS14 scores should be removed from the manuscript.

Additional comments

The paper is well written and timely, and addresses an important topic by investigating the temporal stability of Negative Affect (NA), Social Inhibition (SI) and their combined effect in the Type D personality construct.

1. Please notice that the NA-subscale of the DS14 does not measure “Negative Affect” but rather measures Negative Affectivity”. Negative “affect” refers to a negative mood state, whereas negative “affectivity” refers to the broad and stable tendency to assess negative emotions across time and situations (see Watson & Pennebaker, Psychological Review 1989).

2. Please also notice that nowadays, Type D is written with a capital letter “T” and without a hyphen between “Type” and “D”.

3. I find the conclusion in the abstract somewhat misleading, by stating that “In any case, as the temporal stability of Type D is limited, …”. From my point of view, the temporal stability of the Type D scales over a 6-year period is exactly that what can be expected prototypically from most personality traits. As reported by the author, the correlation between the traits at T1 and T2 was 0.72 for NA and 0.82 for SI. These findings indicate a high level of long-term temporal stability for the both Type D traits. I agree that the stability of the Type D classification across time needs to be improved, but the excellent temporal stability of the Type D components is now not being adequately reflected in the abstract. Moreover, the best predictors of Type D at T2 were the Type D traits assessed 6 years earlier at T1.

4. As correctly pointed out by the author, depression was found to be a better prognostic predictor than Type D personality status in one study (Damen et al., 2013). However, there are also studies were neither depression/anxiety nor Type D predicted mortality (e.g., Pelle et al., 2010). And finally, there are also studies were Type D personality was found to be a better prognostic predictor than depression (e.g., Martens EJ et al., J Clin Psychiatry 2010, 71:778-783).

5. Of the 471 potential participants who initially agreed to participate in a longitudinal study, only 285 eventually did participate in the study, meaning that 40% dropped out of the study. This is a limitation of the study. Moreover, participants in the follow-up study were asked to complete the personality assessment online via computer, and it is unclear how this may have affected the results of the Type D assessment as compared to the baseline assessment.

6. I am afraid it is not appropriate to calculate a total DS14 score. The DS14 assesses two distinctly different personality traits, NA and SI: therefore, scores on these NA and SI measures should not be added as a total DS14 score. Accordingly, it is not appropriate to use cut-off points on the total score of the DS14 to assess the temporal stability of the Type D construct.

7. The Swedish study of cardiovascular patients (Condén E et al., 2014) is highly problematic because the first DS14 assessment of Type D in hospital was done with an interview and not with the standardized self-report measure - as a result, the prevalence of Type D according to the interview in hospital was only 14%. At 1-month follow-up, Type D personality was assessed with the actual DS14 self-report measure, and there the prevalence of Type D was 25% (which is in agreement with the general prevalence of Type D in most previous research in cardiac populations).

---

## Round 0.2 · accepted · Accept

· Academic Editor

Accept

I am pleased to inform you that your manuscript has been judged scientifically suitable for publication. Thank you for your contribution.